# The fall armyworm strain associated with most rice, millet, and pasture infestations in the Western Hemisphere is rare or absent in Ghana and Togo

Rodney N. Nagoshi[1]*, Djima Koffi[2,3], Komi Agboka[3], Anani Kossi Mawuko Adjevi[3], Robert L. Meagher[1], Georg Goergen[4]

1 Center for Medical, Agricultural and Veterinary Entomology, United States Department of Agriculture-Agricultural Research Service, Gainesville, Florida, United States of America, 2 African Regional Postgraduate Programme in Insect Science, University of Ghana, Accra, Ghana, 3 Ecole Supérieure d'Agronomie, Université de Lomé, Lomé, Togo, 4 International Institute of Tropical Agriculture (IITA), Cotonou, Benin

* rodney.nagoshi@usda.gov

**Data Availability Statement:** All relevant data are within the manuscript.

## Abstract

The moth pest fall armyworm, *Spodoptera frugiperda* (J.E. Smith) (Lepidoptera: Noctuidae) is now present throughout much of the Eastern Hemisphere where it poses a significant economic threat to a number of crops. Native to the Western Hemisphere, fall armyworm is one of the primary pests of corn in the Americas and periodically causes significant economic damage to sorghum, millet, cotton, rice, and forage grasses. This broad host range is in part the result of two populations historically designated as host strains (C-strain and R-strain) that differ in their host plant preferences. Reports of infestations in Africa have to date mostly been limited to the C-strain preferred crops of corn and sorghum, with little evidence of an R-strain presence. However, this could reflect a bias in monitoring intensity, with the R-strain perhaps being more prevalent in other crop systems that have not been as routinely examined for the pest. Because knowledge of whether and to what extent both strains are present is critical to assessments of crops at immediate risk, we analyzed specimens obtained from a systematic survey of pasture grass and rice fields, habitats typically preferred by the R-strain, done contemporaneously with collections from corn fields in Ghana and Togo. Substantial larval infestations were only observed in corn, while pheromone trap capture numbers were high only in corn and rice habitats. Little to no fall armyworm were found in the pasture setting. Comparisons with a meta-analysis of studies from South America identified differences in the pattern of strain-specific markers typically found in fall armyworm collected from rice habitats between the two hemispheres. Genetic tests of specimens from rice and corn area traps failed to show evidence of differential mating between strains. These results are consistent with the R-strain being rare or even absent in Africa and, at least for the Ghana-Togo area, this R-strain lack does not appear to be due to limitations in pest monitoring. The implications of these results to the crops at risk in Africa and the accuracy of existing molecular markers of strain identity are discussed.

**Funding:** The author R.N.N. received support came from the Agricultural Research Service of the United States Department of Agriculture (6036-2200-30-00D) and USAID PASA (908-0210-012). The author G.G. received funding from the Deutsche Gesellschaft für Internationale Zusammenarbeit (81235252 GA). The funders had no role in study design, data collection and analysis, decision to publish, or preparation of the manuscript.

**Competing interests:** The authors have declared that no competing interests exist.

## Introduction

Fall armyworm, *Spodoptera frugiperda* (J.E. Smith) (Lepidoptera: Noctuidae), is a moth native to the Western Hemisphere where it is a major pest of corn and several other crops. Its discovery in western Africa in 2016 [1] and subsequent detections in eastern and southern Africa (2017) [2, 3], India (2018) [4], southeastern Asia (2018–2019) [2, 5], and most recently in Australia (2020) [6], presents a threat to agriculture in the Eastern Hemisphere estimated to be in the billions USD with corn (maize) the most impacted [3]. The potential for losses in other crops is significant given fall armyworm behavior in the Western Hemisphere where it is capable of feeding on over 350 different host plants, although consistent economic damage is generally limited to corn, rice, sorghum, millet, soybean, wheat, alfalfa, cotton, turf, and feed grass crops [7, 8].

Major contributors to this broad host range are two populations denoted as "host strains", with the C-strain preferentially impacting corn, sorghum, and cotton while the R-strain predominates in alfalfa, pasture, and forage grasses [9–11]. The host strains are morphologically indistinguishable, with the initial descriptions based on molecular differences found in populations collected from corn and rice, leading to their designation as the "corn strain" and "rice strain" [12]. However, some studies found substantial variability in the strain markers found in collections from rice, particularly when compared to other rice-strain preferred hosts such as pasture and turf grasses [13, 14]. These observations suggest that rice is probably not the primary host for this fall armyworm group and brings into question its strain specificity. For this reason, we have taken to designating the corn and rice strains as the C-strain and R-strain, respectively, but note that both sets of terms are found in the literature.

The taxonomic and genetic status of the two fall armyworm host strains remain uncertain and controversial. Since their original designation as strains [12, 15], the characterizations of the groups have ranged from "host forms" where the divergence is relatively small [13], to "sibling species" associated with evidence of significant reproductive isolation [16]. Genomic studies have been similarly inconclusive with reports for both the presence [17] and absence [18] of significant nuclear genomic differences between strains. Molecular markers remain the best definer of strain identity, as empirically demonstrated by strong correlations between genetic polymorphisms and host plant in fall armyworm surveyed from multiple locations in the Western Hemisphere [14, 19, 20].

The strain-specifying genetic markers include segments of the mitochondrial *Cytochrome Oxidase Subunit I* (*COI*) gene and the Z-chromosome *Tpi* gene that encodes for the housekeeping enzyme Triosephosphate isomerase [19, 21, 22]. In both cases, single base polymorphisms in the coding regions distinguish the strains, with C-strain denoted by *COI*-CS and TpiC, and the R-strain by *COI*-RS and TpiR. Being on different genetic elements, the correspondence between the *COI* and *Tpi* strain haplotypes can be lost with mating between strains. This is generally not the case in Western Hemisphere populations where agreement between the *COI* and *Tpi* strain markers approximates 80% [19], indicating a strong bias for productive mating within strains. This is supported by more detailed analysis of the *Tpi* markers. In the Lepidoptera ZW/ZZ sex determination system fall armyworm males carry two copies of the *Tpi* gene, making it possible to obtain TpiC/TpiR heterozygotes indicative of potential interstrain hybrids [19, 23]. These have been found in Western Hemisphere field populations, though at frequencies significantly lower than expected by allele frequencies, which is suggestive of limited mating between strains [23]. These observations are consistent with laboratory studies describing partial but significant barriers to interstrain hybridization [24–26].

The fall armyworm populations found in the Eastern Hemisphere are unusual in that the great majority in most locations show disagreement between the *COI* and *Tpi* strain

haplotypes. Specifically, populations in eastern and southern Africa [27], India [28], and Myanmar [2] are primarily of the *COI*-RS TpiC configuration, making unclear their strain identification. Because virtually all the Eastern Hemisphere specimens tested to date were collected from the C-strain associated hosts corn and sorghum, it appears that these populations are behaviorally of the C-strain and *COI* is no longer an indicator of fall armyworm plant host preference. In contrast, the observed predominance of TpiC is consistent with the collection data and with *Tpi* remaining an accurate marker of strain identity. However, an important caveat is that these observations are so far limited to fall armyworm collected from C-strain preferred host plants. The strain marker composition of specimens using R-strain hosts in Africa is currently undetermined.

The apparent rarity of the R-strain in the Eastern Hemisphere as indicated by both the near absence of TpiR [2, 27–29] and the infrequency of reports of fall armyworm infestations outside of corn and sorghum, suggests that R-strain preferred crops are at low risk for this pest in Africa and Asia. However, it is possible that these observations reflect inadequate monitoring for this pest, with significant populations of the R-strain present but undetected in habitats such as pasture and rice fields that may not be surveyed for fall armyworm as intensely as corn. In the absence of more extensive surveys and collections from R-strain associated host plants the possible presence of the R-strain in Africa remains uncertain.

To address these issues, we obtained specimens from systematic surveys of fall armyworm from pasture, rice fields, and corn fields located in the western African countries of Ghana and Togo [30, 31]. Because many specimens were collected from pheromone traps in rice, we performed a meta-analysis of genetic studies of fall armyworm collected from rice in South America to ascertain the strain and genetic marker composition we should expect for this host plant. The Ghana-Togo collections were then analyzed and compared for their distributions of the *COI* and *Tpi* markers. The results were used to assess the frequency of the R-strain at the different surveyed locations. The implications of the data on the risk to R-strain preferred crops such as rice, millet, and pasture grasses in Africa are discussed.

## Materials and methods

Fall armyworm individuals were collected from multiple sites and time periods in Ghana and Togo (Fig 1). Specimens from 2016 and 2017 were limited to larval collections from corn and were previously described (Table 1). Collections from October 2018 to January 2019 were from pheromone traps in habitats represented by corn, rice, or pasture grasses [30, 31]. Attempts were made throughout the 2018–9 survey period to collect larvae from rice and pasture grasses, but observed infestations were rare and only a small number could be processed for genetic analysis. All specimens were stored refrigerated or air dried at ambient temperature until transport by mail to CMAVE, Gainesville, FL USA for DNA preparation. Fall armyworm data from Florida and Texas were derived from collections previously used to study strain mating behavior in the field [23, 32].

### DNA preparation and PCR amplification

DNA from individual specimens were isolated as previously described with minor modifications [32]. In brief, specimens were homogenized in 1.5 ml of phosphate buffered saline (PBS, 20 mM sodium phosphate, 150 mM NaCl, pH 8.0) using a tissue homogenizer (PRO Scientific Inc., Oxford, CT) or hand-held Dounce homogenizer then pelleted by centrifugation at 6000 g for 5 min at room temperature. The pellet was resuspended in 800 μl Genomic Lysis buffer (Zymo Research, Orange, CA) by gentle vortexing and incubated at 55˚C for 15 min, followed by centrifugation at 10,000 rpm for 5 min. The supernatant was transferred to

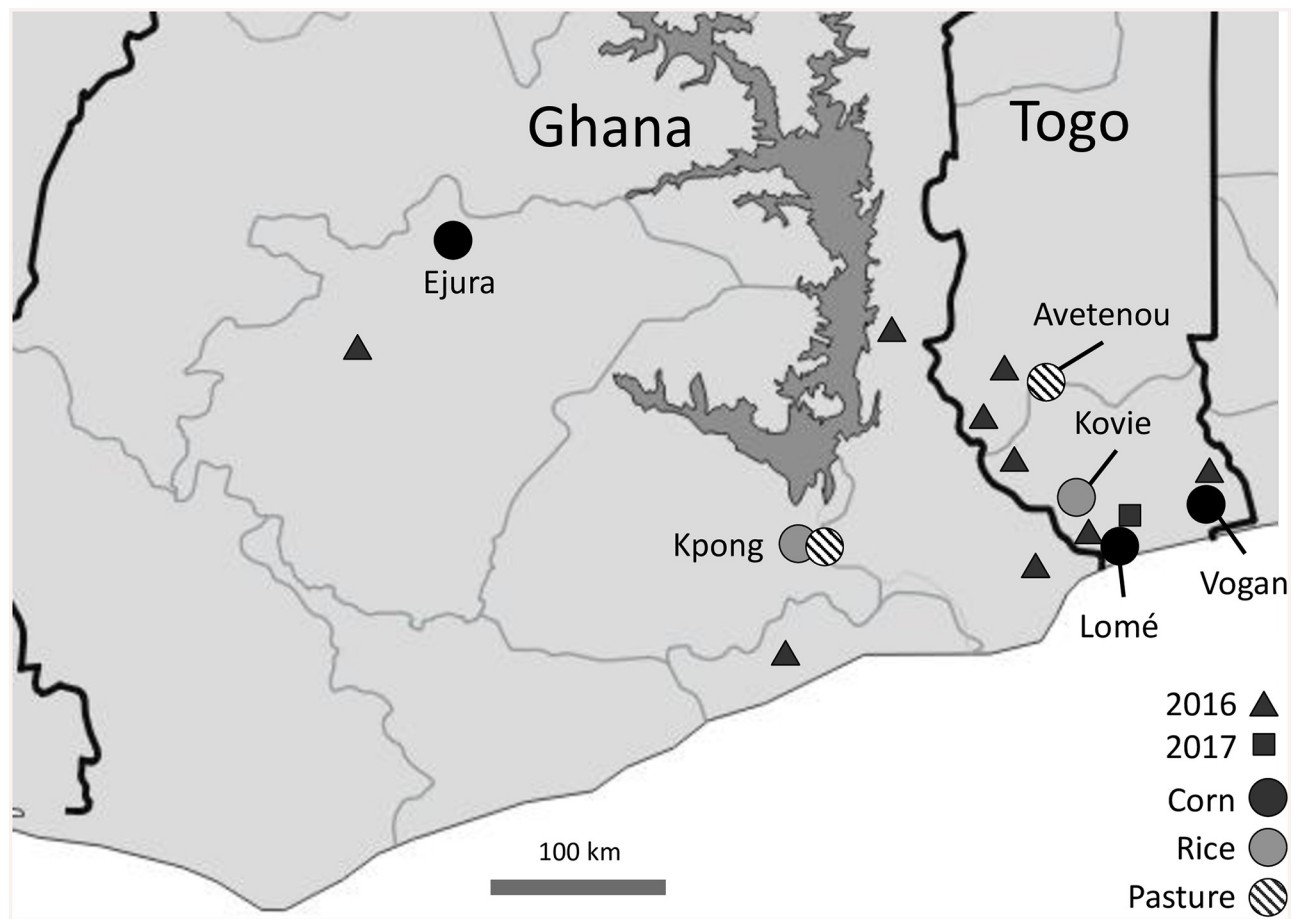

**Fig 1. Map showing locations of fall armyworm collections in Ghana and Togo.** Major towns near locations surveyed in 2018 are labeled. Sites are described in Table 1. The map was generated using QGIS version 2.18.2 (Open Source Geospatial Foundation).

**Table 1. Information for fall armyworm collections.**

| Date | Country | Nearest town | Host | Type | Reference |
|---|---|---|---|---|---|
| Sep 2016 | Ghana | Multiple | Corn | Larval | [27] |
| Oct-Jan 2018–9 | Ghana | Ejura | Corn | Trap | [31] |
| Oct-Jan 2018–9 | Ghana | Kpong | Rice | Trap | [31] |
| Oct-Jan 2018–9 | Ghana | Kpong | Pasture | Trap | [31] |
| Jul-Oct 2016 | Togo | Multiple | Corn | Larval | [33] |
| May-Aug 2017 | Togo | Lomé | Corn | Trap | [28] |
| Oct-Jan 2018–9 | Togo | Vogan | Corn | Trap | [31] |
| Oct-Jan 2018–9 | Togo | Kovie | Rice | Trap | [31] |
| Oct-Jan 2018–9 | Togo | Avetenou | Pasture | Trap | [31] |
| Sep 2005 | Florida, USA | Hague | Corn | Trap | [34] |
| Jan-Feb 2012 | Florida, USA | Orlando | Corn | Trap | [34] |
| Nov-Dec 2006 | Texas, USA | Corpus Christi | Corn | Trap | [32] |
| Jan-Apr 2012 | Texas, USA | Weslaco | Corn | Trap | [32] |

a Zymo-Spin III column (Zymo Research, Orange, CA) and processed according to manu-facturer's instructions.

Polymerase chain reaction (PCR) amplification was performed using a 30-μl reaction mix containing 3 μl of 10X manufacturer's reaction buffer, 1 μl 10 mM dNTP, 0.5 μl 20-μM primer mix, 1 μl DNA template (between 0.05–0.5 μg), 0.5 units Taq DNA polymerase (New England Biolabs, Beverly, MA) with the remaining volume water. The thermocycling program was 94˚C (1 min), followed by 33 cycles of 92˚C (30 s), 56˚C (45 s), 72˚C (45 s), and a final segment of 72˚C for 3 min. Amplification of *COI* used the primer pair *COI-891F* (5'-TACACGAGCA TATTTTACATC-3') and *COI-1472R* (5'-GCTGGTGGTAAATTTTGATATC-3') to produce a 603-bp fragment. Amplification of the *Tpi* region was done with the primers *Tpi412F* (5'-CCGGACTGAAGGTTATCGCTTG -3') and *Tpi1140R* (5'-GCGGAAGCATTCGCTGACAACC -3') that spans a variable length intron to produce a fragment with a mean length of 500 bp. Primers were synthesized by Integrated DNA Technologies (Coralville, IA). Gel electrophore-sis and fragment isolation were done as previously described [32]. DNA sequencing was per-formed directly from the gel purified PCR fragments by Sanger sequencing, using primers *COI-924F* or *Tpi412F* (Genewiz, South Plainfield, NJ).

The specimens were of variable quality and in many cases a single PCR amplification did not produce sufficient product. In these cases, a nested PCR protocol was performed. For COIB analysis, the first PCR amplification was performed with the primer pair *COI-891F* and *COI-1472R*. One microliter of this first reaction was amplified using primers *COI-924F* (5'-TTATTGCTGTACCAACAGG-3') and *COI-1303R* (5'-CAGGATARTCAGAATATCGACG-3'). For the *Tpi* marker, the first amplification was performed using primers *Tpi469F* (5'-AAGGACATCGGAGCCAACTG-3') and *Tpi1195R* (5'-AGTCACTGACCCACCATACTG-3'). One microliter of the first reaction was then amplified using primers *Tpi412F* and *Tpi1140R*. Relative locations of the primers are described in Fig 2.

## Determination of strain-identity using *COI* and *Tpi*

The diagnostic genetic markers that distinguish the C-strain and R-strain are single nucleotide polymorphisms typically associated with neutral substitutions (Fig 2A and 2B). *COI* and *Tpi* gene sites are preceded by an "m" (mitochondria) or "g" (genomic), respectively. This is fol-lowed by the gene name, number of base pairs from the predicted translational start site for *COI*, or the 5' start of the exon for *Tpi*. The observation of multiple nucleotides possible at a given position is described using IUPAC convention (R: A or G, Y: C or T, W: A or T, K: G or T, S: C or G, D: A or G or T).

To facilitate the screening of large number of samples, strain identity was defined by a sin-gle site in *COI* and *Tpi*, though the accuracy of this determination was continually checked by comparisons with nearby strain-specific SNPs. Strain identity defined by COIB, a segment of the *COI* gene lying near the 3' end, was determined by SNP mCOI1164, with an A or G signify-ing C-strain and a T indicating R-strain (Fig 2A). Sites mCOI1176 and mCOI1182 show a sim-ilar polymorphism pattern as mCOI1164. Strain identity by *Tpi* is defined SNP gTpi183 found in the fourth exon of the predicted *Tpi* coding region (Fig 2B). The nearby SNP, gTpi168 shows the same pattern as gTpi183. The gTpi180 polymorphism segregates with gTpi168 and gTpi183 in African fall armyworm populations, but this site is only infrequently polymorphic and not strain-specific in Western Hemisphere populations [27].

To more accurately calculate *Tpi* haplotype frequency, the TpiH data were added to the TpiC and TpiR counts as follows. In collections from pheromone traps, all specimens are male and therefore carry two copies of the *Tpi* gene, with TpiH carrying one copy each of TpiC and TpiR. In these collections the number of the TpiC haplotype was calculated by 2(number of

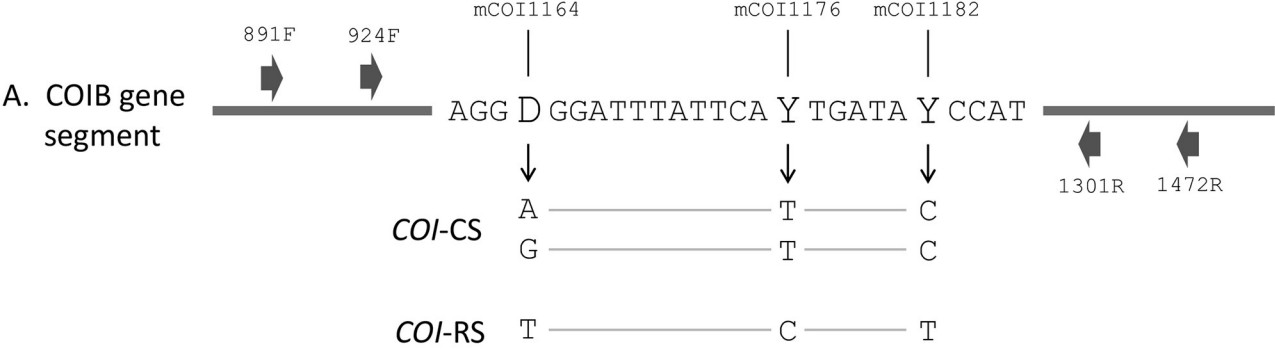

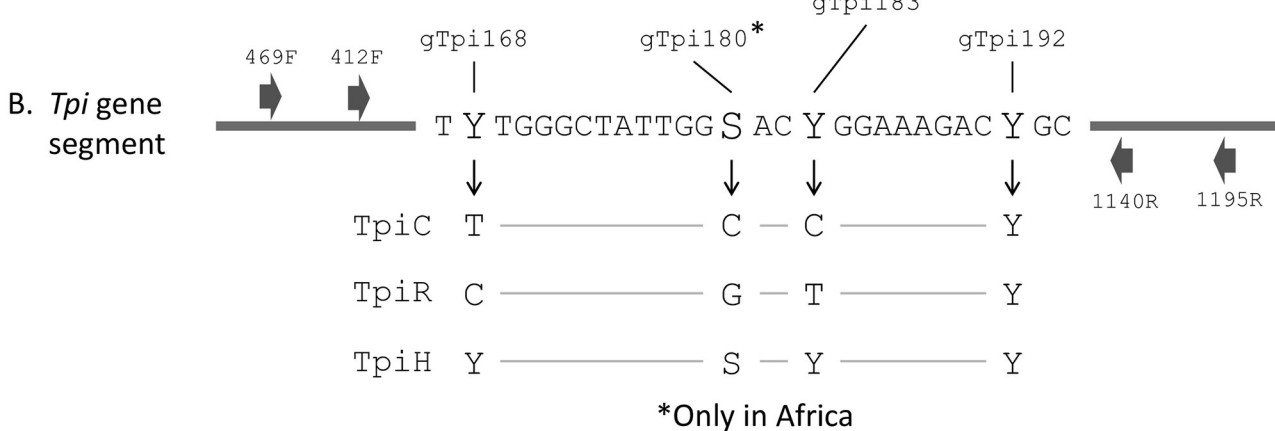

*Only in Africa

**Fig 2. Schematic of gene segments used for the genetic analysis.** A. COIB segment of the mitochondrial *COI* gene showing location of the strain diagnostic mCOI1164 SNP as well as two additional polymorphic sites showing the same strain-specificity. Nucleotides observed at each SNP are listed below arrows and the configurations associated with *COI*-CS and *COI*-RS described. B. Segment of the nuclear *Tpi* gene showing location of the gTpi183 site that is diagnostic of strain identity in Western Hemisphere populations. Site gTpi168 shows a similar polymorphic distribution as gTpi183. The gTpi180 polymorphism is not typically found in the Western Hemisphere but is present in Africa. Nucleotides observed at each site are listed below arrows and the configuration associated with TpiC and TpiR indicated. Because *Tpi* is Z-linked, males have two copies of the gene and so can be heterozygous for these polymorphisms (TpiH). Site gTpi192 is polymorphic for C or T in both strains (strain nonspecific). Nomenclature follows IUPAC convention where Y = C or T; S = C or G; R = A or G; and D = A, G, or T. Small block arrows denote location of relevant primers used for PCR and DNA sequencing.

TpiC specimens) + (number of TpiH specimens) and the TpiR haplotype by 2(TpiR) + (TpiH). In the larval collections gender was typically not identified. In this case a 1:1 sex ratio was assumed with half the collection considered male. In these collections the number of *Tpi* haplotypes was calculated as 1.5(TpiC) + (TpiH) or 1.5(TpiR) + (TpiH).

## Testing for strain-specific mating behavior in field populations

The segment of the *Tpi* gene used to identify strain also carries a SNP, gTpi192, that shows much less strain-specificity than the nearby strain-diagnostic gTpi183 site (Fig 2B). These were used to develop a method for assessing strain-specific mating behavior in field populations [23]. Productive matings within strains (intrastrain mating, C X C and R X R) occurs at a much higher frequency than mating between strains (interstrain mating, C X R and R X C) [19, 24–26]. A consequence of this strain-specific mating bias is that heterozygosity at the

strain-specific site gTpi183 is significantly reduced relative to the less specific gTpi192 site, which can be compared using the inbreeding coefficient $F_{IS}$ [23, 32]. Heterozygotes at each site is detected by examination of the DNA sequence chromatograph curves. Specifically, both gTpi183 and gTpi192 are polymorphic for C and T so an overlapping C and T profile is indicative of heterozygosity.

The frequencies of the C-allele and T-allele were estimated using Hardy-Weinberg equilibrium analysis. The allele frequencies for C and T are given by $p$ and $q$, respectively, such that $p + q = 1$, with $p$ calculated by the equation $p = \text{freqCC} + 0.5[\text{freqY}]$, where freqCC is the observed frequency of CC homozygotes and freqY the observed frequency of Y (CT) heterozygotes. The frequency of the T-allele ($q$) is then given by the equation $q = 1 - p$. The local expected heterozygote frequency, $H_e$, is equal to the equation $H_e = 2pq$. The local observed heterozygote frequency, $H_o$, is given by the empirically determined freqY (the frequency of overlapping chromatograph curves). Wright's local inbreeding coefficient, $F = (H_e - H_o)/H_e$, was calculated for SNPs gTpi183 and gTpi192.

All collections used to study interstrain mating frequencies were derived from pheromone trap captures. This means that all specimens are adult males and therefore carry two copies of the *Tpi* gene.

## DNA sequence and statistical analysis

DNA sequence alignments and comparisons were performed using programs available on the Geneious 10.0.7 software (Biomatters, Auckland, New Zealand). Basic mathematical calculations and generation of graphs were done using Excel and PowerPoint (Microsoft, Redmond, WA). Other statistical analyses including ANOVA and *t*-tests were performed using GraphPad Prism version 9.1.0 for Mac (GraphPad Software, La Jolla California USA). ANOVA calculations were combined with Tukey multiple comparisons testing to make pair-wise comparisons.

## Results

### Fall armyworm collections from different host plants

A systematic effort was made from October 2018 to January 2019 to collect fall armyworm from corn, rice and pasture grass habitats by pheromone trapping [26]. Substantial numbers were collected from the corn and rice traps, but not in the pasture habitat (Fig 3A and 3B). At the Ghana sites, captures were highest in the corn habitat, peaking in mid-December, while the rice site captures were observed by late October and remained unchanged through December (Fig 3A). At the Togo sites, captures were highest in the rice habitat and were still rising at the end of December while captures in the corn fields peaked in mid-November (Fig 3B).

### Higher frequency of R-strain markers in South American rice infestations

Rice is a major food source for many parts of the world that are now threatened by fall armyworm. To better assess the risk to rice posed by this pest, we performed a meta-analysis of fall armyworm strain distributions in rice crops in South America, where most published studies of this type have been done. Juarez et al (2012) found substantial variations in strain composition in collections from rice as determined by the COIB marker, with the frequency of the R-strain *COI*-RS marker ranging from 0 to 100% ([9], Fig 4A). More consistently high *COI*-RS frequencies were reported in two other studies examining rice collections from Brazil [35] and Argentina [14]. Similar variability was found in collections from corn, but with a lower average frequency. Overall, 63% (5/8) of the rice collections showed a *COI*-RS majority compared to

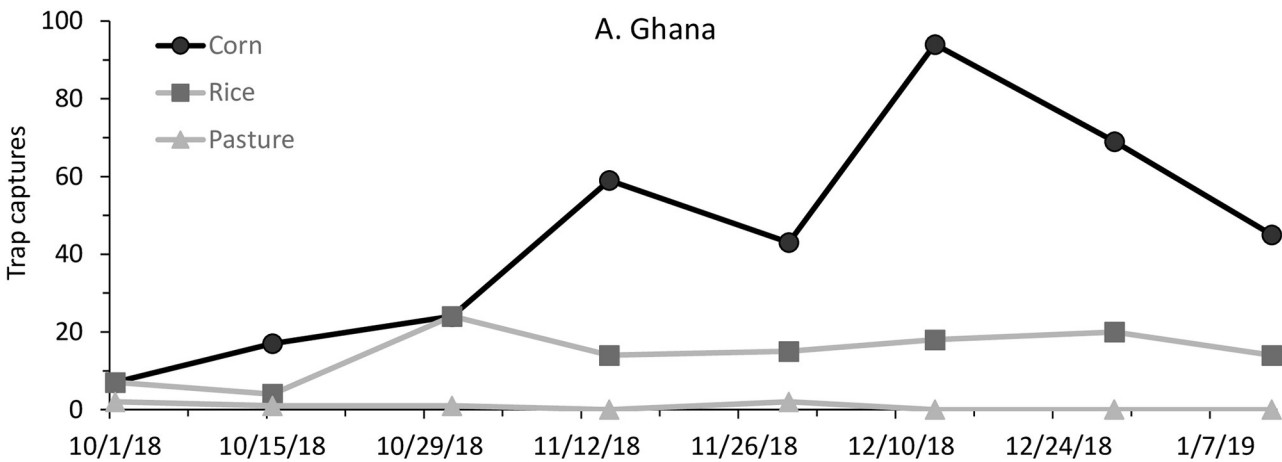

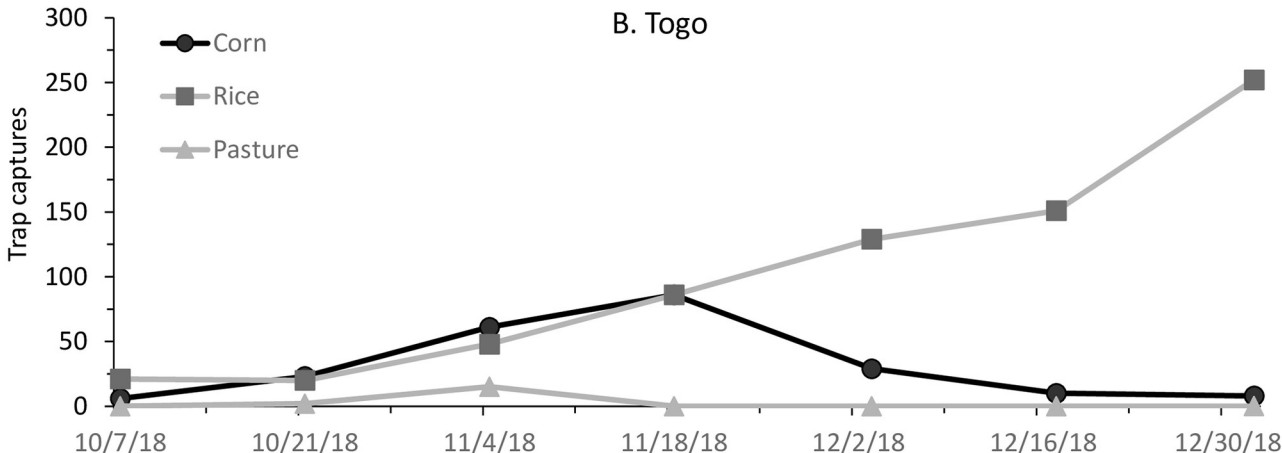

**Fig 3. Graph of fall armyworm pheromone trap captures in corn, pasture, and rice habitats in Ghana (A) and Togo (B).**

21% (5/24) of corn collections. Despite the high variability the mean *COI*-RS frequency was statistically different between the two host crops, with the R-strain marker more prevalent in fall armyworm from rice hosts than corn ($P = 0.0405$, $t = 2.141$, $df = 30$). These findings are consistent with those using the TpiR marker for these collections (Fig 4B). The R-strain TpiR marker was in the majority in 67% of collections from rice versus 6% from corn, with a statistically significant difference in mean frequency ($P = 0.0013$, $t = 3.731$, $df = 20$). These data indicate that in South American populations, the R-strain as defined by the *COI* and *Tpi* molecular markers does prefer rice over corn hosts in the field, though it can be found in both plant species at highly variable frequencies.

## The R-strain markers are not preferentially found in Ghana-Togo rice fields

The *COI*-RS marker was in the minority in all the rice collections and was similarly distributed in corn where only one of six collections had a *COI*-RS majority (Fig 5A). The difference in mean *COI*-RS frequencies for the three groups (Ghana-Togo rice, Ghana-Togo corn, Florida-

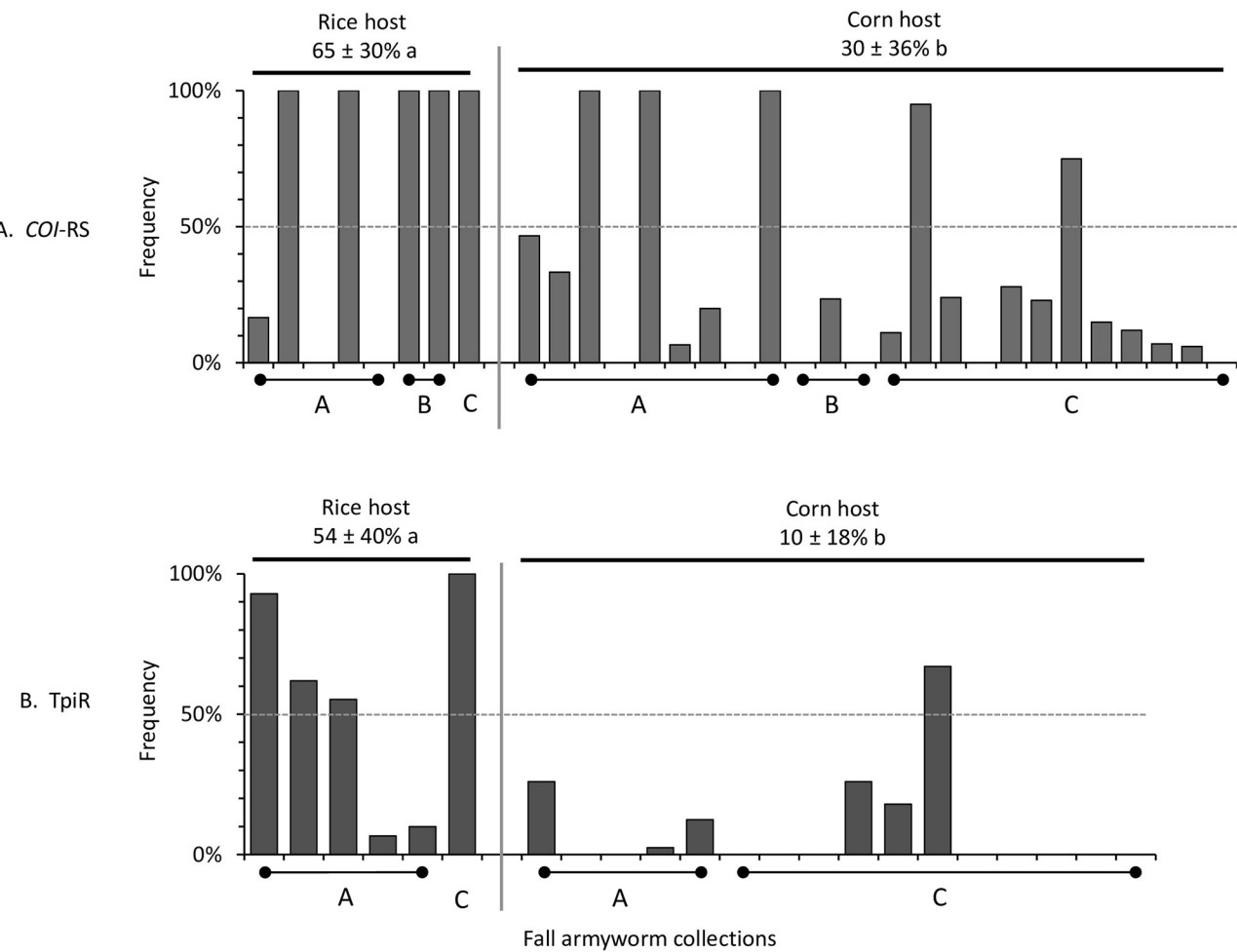

**Fig 4. Bar graphs showing distribution of the R-strain markers *COI*-RS (A) and TpiR (B) in rice and corn habitats in Argentina and Brazil.** Data are from three studies, Juarez et al. (2012) [9], Machado et al. (2007) [35], and Murua et al. (2015) [14]. Mean frequencies (± standard deviation) are noted above columns with different letters indicating statistically significant differences using a two-tailed t-test.

Texas corn) were not significant by ANOVA analysis ($P = 0.3892$, $r^2 = 0.3144$, $F = 1.146$), indicating no specific association of this marker with host plant.

In all locations in Ghana and Togo, most of the TpiR haplotype was heterozygous with TpiC (TpiH), which made up 13% of the collected specimens compared to less than 2% that were either hemizygous or homozygous for TpiR (Fig 5B). This contrasts with Florida and Texas collections from corn where TpiR hemizygotes/homozygotes outnumbered TpiH, with a mean TpiR frequency of 31% (Fig 5B). ANOVA analysis of the mean frequencies of the TpiR haplotype as calculated from the TpiR and TpiH numbers indicated a significant difference between the Ghana-Togo and Florida-Texas collections ($P = 0.0003$, $r^2 = 0.9634$, $F = 65.78$). Specifically, significant differences were observed between the mean TpiR frequency found in Florida-Texas corn versus both Ghana-Togo rice ($P = 0.009$) and Ghana-Togo corn ($P = 0.002$), but not between the Ghana-Togo rice and corn collections ($P = 0.3485$). These findings are consistent with previous observations that TpiR frequencies at corn sites in Africa are consistently lower than what is typical for Western Hemisphere corn site collections [22, 30].

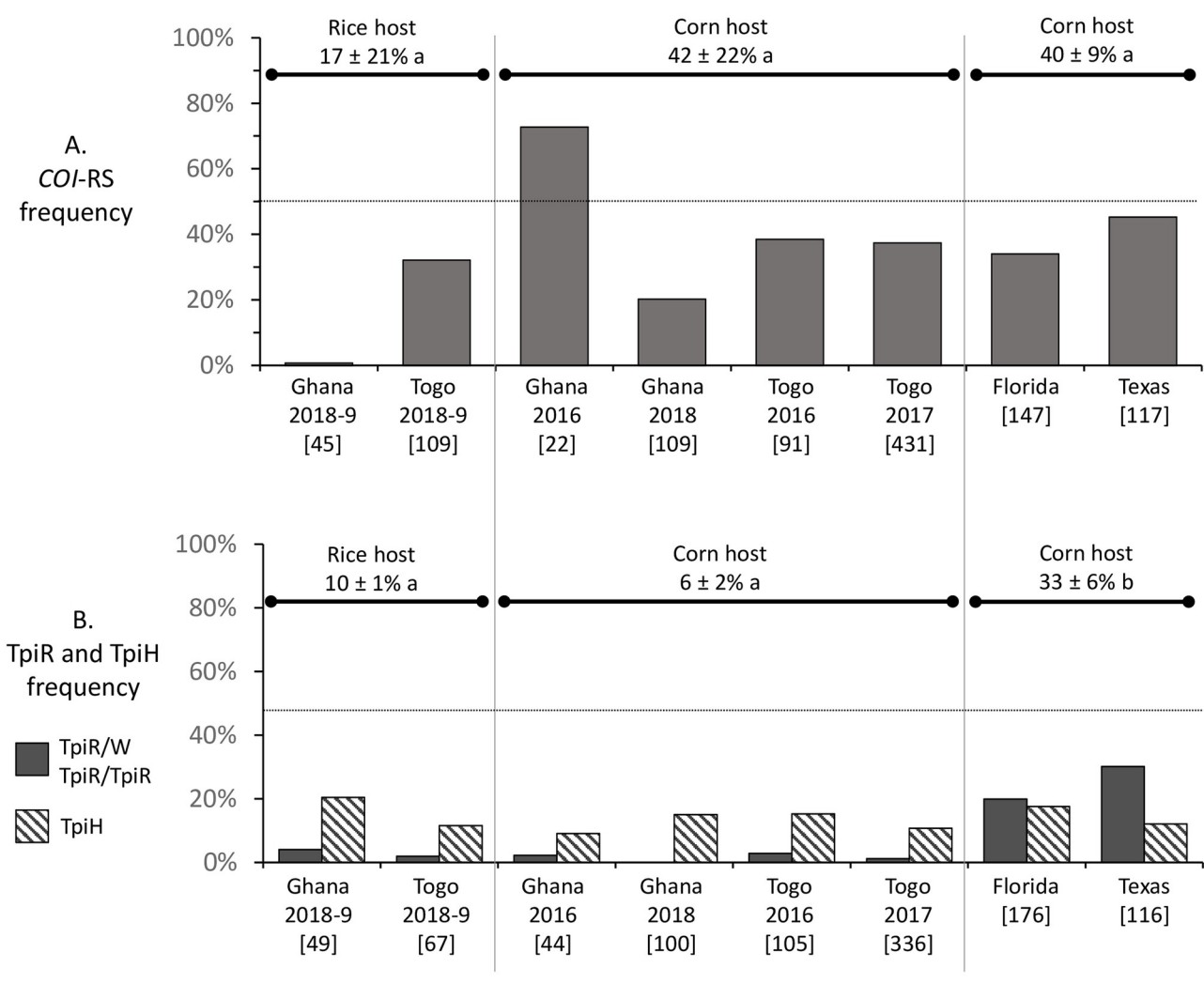

**Fig 5. Bar graphs showing distribution of the R-strain markers *COI*-RS (A) and TpiR (B) in rice and corn habitats in Ghana and Togo.** Collections are as described in Table 1. Mean haplotype frequencies (± standard deviation) are noted above columns. Within each graph, frequencies with different lower-case letters are statistically different. Numbers in brackets indicate specimens tested from each collection. For TpiR, the graphs show the frequencies of TpiR hemizygotes or homozygotes (dark fill) and TpiH heterozygotes (diagonal lines). TpiR haplotype frequencies were calculated by combining these classes as described in the Methods.

## No significant fall armyworm infestations in rice or pasture

Despite the high level of trap collections in the rice habitat, very few larvae were detected infesting rice plants. Five larval specimens collected from rice were examined. Four were identified as *Spodoptera littoralis* (Boisduval) (Lepidoptera: Noctuidae) based on the COIB DNA sequence, while one displayed the fall armyworm *COI*-RS and TpiC markers. Trap captures in the pasture habitat were rare and no larvae were detected. We obtained genetic data from two specimens and both expressed the *COI*-CS TpiC markers. Larvae were also found in a nearby field of cabbage, an atypical fall armyworm host. Of the eight specimens examined one appeared to be *S. littoralis*, three were *COI*-CS TpiC, and four were *COI*-RS TpiC.

## Fall armyworm in rice habitats do not display strain-specific mating patterns

To test whether the TpiR moths in the rice collections are behaving in a manner consistent with the R-strain, we applied a strategy that measures the degree of differential mating between strains (as identified by *Tpi*) in field populations [23]. Because the polymorphisms at the strain-specific gTpi183 SNP are asymmetrically distributed between strains, heterozygotes at this site arise primarily from interstrain matings. In comparison, heterozygotes at the nearby nonspecific gTpi192 SNP can occur by either interstrain or intrastrain hybridization. Since laboratory studies indicate that mating between strains is reduced relative to intrastrain hybridization [24, 26], the frequency of heterozygotes at gTpi183 in field populations should be lower than expected from random mating and lower than observed for gTpi192. In summary, the difference in heterozygosity between gTpi183 and gTpi192 is a measure of the effect of strains on hybridization, with gTpi192 acting as an internal control for factors unrelated to strain mating behavior.

The detection of differential mating is exemplified in the pheromone trap collections from Florida and Texas. The frequency of heterozygotes was measured by the inbreeding coefficient $F_{IS}$, a metric that compares the heterozygote frequency observed with that expected from the allele frequencies and the assumption of random mating. The $F_{IS}$ metric approaches +1.0 when heterozygotes occur less frequently than expected (observed < expected), nears -1.0 when the converse is true (observed > expected), and ranges around 0.0 when mating is random (observed = expected). The four collections from Florida and Texas consistently show for the strain-specific gTpi183 site an $F_{IS}$ greater than 0.5 with a mean of 0.67, which was significantly different from the mean $F_{IS}$ of 0.07 for the nonspecific gTpi192 site (two-tailed paired *t* test, *P* = 0.0016, *t* = 10.99, *df* = 3) (Fig 6). The differences between the gTpi183 and gTpi192 $F_{IS}$ values (designated δ) ranged from 0.48 to 0.78 with a mean of 0.59. These results are consistent with interstrain mating barriers limiting gTpi183 heterozygote formation while normal levels of intrastrain mating produce frequencies of gTpi192 heterozygotes at levels approximating that expected from random mating.

In comparison the mean δ for the four Ghana and Togo pheromone trap collections was 0.03, indicating no differential mating between TpiR and TpiC fall armyworm (Fig 6). However, a difference between the rice and corn collections was observed, as both rice collections displayed positive δ values of 0.26 (Ghana) and 0.15 (Togo) compared to the negative δ values (-0.17 and -0.19 for the Ghana and Togo, respectively) found in the corn collections. These are much lower than the mean δ for Florida and Texas (0.59). Overall, there was no significant difference between mean gTpi183 and gTpi192 $F_{IS}$ values (two-tailed t-test, *P* = 0.5893, *t* = 0.5701, *df* = 6) for the Ghana and Togo collections, while a significant difference was observed for δ between the Ghana-Togo and Florida-Texas collections (two-tailed unpaired *t* test, *P* = 0.007a, *t* = 3.963, *df* = 6).

## Discussion

The presence of the fall armyworm R-strain in Africa would have significant consequences to agriculture in the affected areas by increasing the range of crops at risk of substantial and consistent infestations. To further investigate this issue, we made a concerted effort to obtain specimens from expected R-strain host plants for genetic analysis. Difficulties in finding larval infestations in rice and pasture grasses were compensated for with pheromone traps, a method previously shown to be far more efficient at collecting fall armyworm while still capable of revealing strain differences in habitat distribution [10, 14, 36, 37].

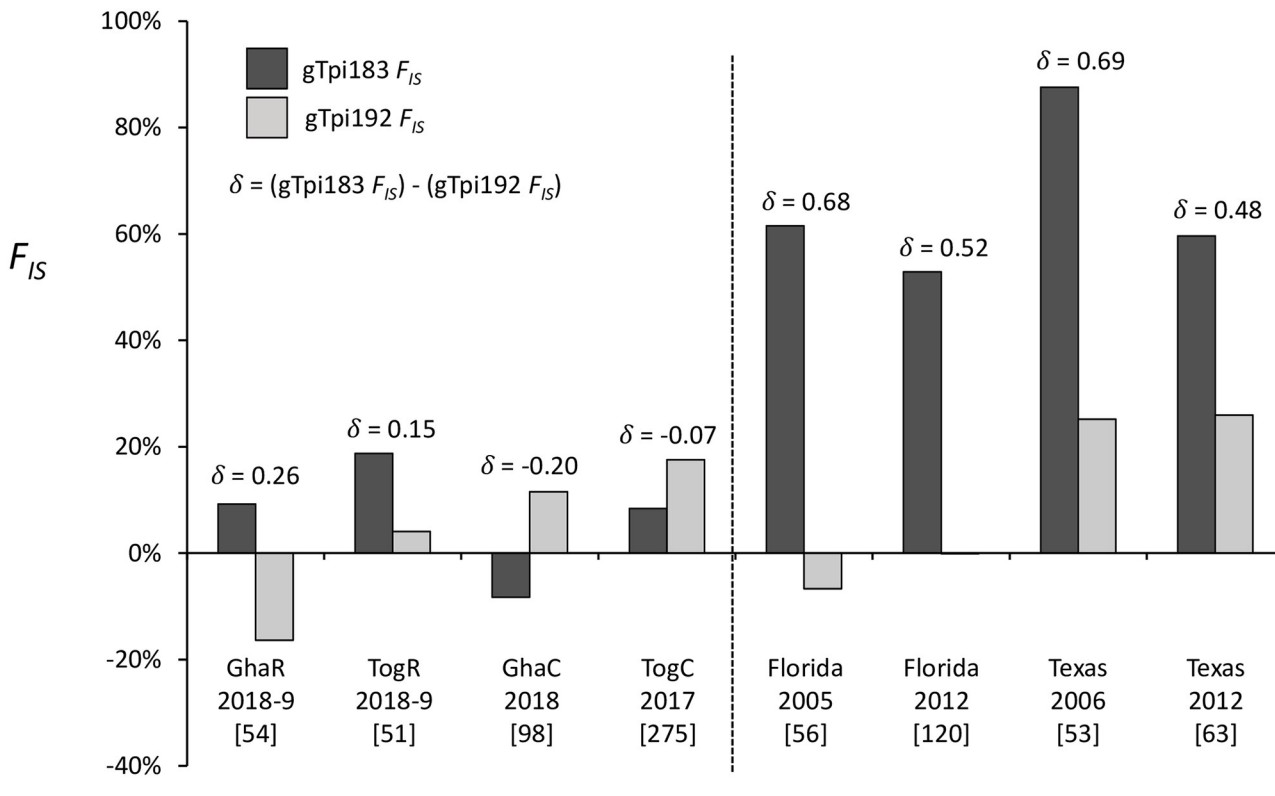

**Fig 6. Bar graph of the inbreeding coefficient, $F_{IS}$, calculated for different fall armyworm collections.** The number above each column pair denotes the difference ($\delta$) between the $F_{IS}$ values, $\delta$ = (gTpi183 $F_{IS}$)–(gTpi192 $F_{IS}$). Numbers in brackets indicate specimens tested from each collection.

Despite these efforts, very few moths were detected in pasture settings (Fig 3), which in the Western hemisphere is a major source of the R-strain [10]. In contrast, substantial numbers of fall armyworm were collected from pheromone traps placed in rice sites in Ghana and especially Togo during most of the survey period. However, very few larvae in rice were detected at both locations. This non-correspondence between trap numbers and local infestations suggests that the rice trap captures may have originated from other hosts. In both Ghana and Togo, rice is grown in lowlands where adequate water for most of the year allows multiple crops to be grown contemporaneously and in proximity. Because the farms in this western African area are usually small, less than two hectares in size, the rice habitat collection sites were unavoidably located within 1–2 kilometers of corn and other crops [30, 31]. These factors make it possible, if not likely, that fall armyworm collected in the rice field pheromone traps originated from nearby corn or other hosts. A similar lack of fall armyworm infestations in rice was observed by one of us (G. G.) who participated in systematic surveys of a large array of rice varieties grown on the campus of IITA Cotonou, Benin and found no fall armyworm larvae despite frequent infestations in other crops in the area. Regardless of origin, fall armyworm found in rice traps were similar to those found elsewhere in being predominantly of the TpiC haplotype. In summary, we found no evidence for the R-strain either in terms of TpiR frequency or observations of significant fall armyworm infestations in R-strain preferred pasture or rice host plants.

The disagreement between the *COI* and *Tpi* strain haplotypes found in most African fall armyworm populations could be explained by interstrain hybridization [32]. Specifically, an

R-strain female crossed to a C-strain male (R X C) will produce female progeny that carry the *COI*-RS mitochondria haplotype and are hemizygous for TpiC (TpiC/W). The mating of these hybrid females to C-strain males will then produce a stable *COI*-RS TpiC lineage. We believe it likely that the association of *Tpi* with strain identity is because it is linked to one or more genes driving divergence between strains, a supposition supported by the mapping to the *Z*-chromosome of a locus that confers partial sterility in interstrain hybrids [25]. If correct, then even after crosses leading to the disassociation of the *COI* haplotyped from strain identity the TpiC marker would still be linked to functions that define the C-strain and thereby remain an accurate strain marker.

The genetic studies to date indicate that the described set of interstrain crosses is plausible. There is evidence that the R X C cross does occur in field populations and at a higher frequency than that of the reciprocal mating [19, 20]. Laboratory studies show that the resultant hybrid females can productively mate with C-strain males, though with much reduced fertility [24, 25]. The fertility of subsequent generations is unknown but is presumably high given the predominance of this genotype in Africa. Why this hybrid lineage should have expanded in Africa is not known, but it was previously hypothesized that this could be due to the unusual circumstances faced by an invasive propagule [27]. Genetic admixture is known to occur when previously separated populations of the same species are introduced into a novel habitat [38, 39], which in the case of fall armyworm would mean higher rates of interstrain hybridization. In addition, a small initial population can lead to inbreeding depression within strains [38], thereby leading to fitness advantages for the interstrain hybrids. While clearly speculative, this scenario provides an explanation for discordance between the *COI* and *Tpi* strain markers and the continued preference of these hybrids to C-strain host plants.

We reasoned that if the *Tpi* haplotypes are still accurate markers of strain identity in Africa, then the relative frequencies of TpiC, TpiR, and TpiH should reveal evidence of differential mating similar to that observed in the Western Hemisphere [23]. Unfortunately, the results of this analysis were variable and inconclusive, with a suggestion of differential mating in the rice collections that was much lower in magnitude than that observed in Florida-Texas and not confirmed by the collections in Ghana-Togo corn (Fig 6). Potentially complicating this analysis is that the frequency of TpiR homozygotes in Ghana and Togo is very low, making up less than 2% (11/686) of the sampled specimens. Furthermore, of the few males carrying TpiR, most were heterozygous with TpiC (TpiH, Fig 5), a potentially hybrid genotype where the expected mating behavior is unknown. This contrasts with the Florida-Texas populations from corn where 24% (70/292) of the collection were homozygous TpiR compared to 15% (45/292) TpiH. Therefore, we think it possible that the frequency of TpiR homozygous males in the Ghana-Togo collections may be too low to consistently detect differential mating between strains by this method.

In conclusion, the results to date indicate that the fall armyworm R-strain is not (yet) present in significant numbers in Africa, reducing the immediate risk of economic damage to rice, millet, pasture, and forage grasses. The *COI* marker continues to be disassociated from strain identity in African fall armyworm populations while the status of the *Tpi* markers is uncertain and likely to remain so until the occurrence of either TpiR or infestations in R-strain associated host plants increase to levels that allow for conclusive testing. However, given the correspondence of the TpiR marker to the R-strain in Western Hemisphere populations, the continued surveillance for TpiR in the Eastern Hemisphere is recommended as the introduction or augmentation of the R-strain by invasion could greatly exacerbate the economic damage caused by this species.

## Acknowledgments

We recognize Dr. J.M.G. Thomas for technical assistance in preparing the specimens. The use of trade, firm, or corporation names in this publication is for the information and convenience of the reader. Such use does not constitute an official endorsement or approval by the United States Department of Agriculture or the Agricultural Research Service of any product or service to the exclusion of others that may be suitable.

## Author Contributions

**Conceptualization:** Rodney N. Nagoshi, Djima Koffi.

**Formal analysis:** Rodney N. Nagoshi.

**Funding acquisition:** Rodney N. Nagoshi, Georg Goergen.

**Investigation:** Rodney N. Nagoshi, Djima Koffi, Komi Agboka, Anani Kossi Mawuko Adjevi, Robert L. Meagher, Georg Goergen.

**Methodology:** Rodney N. Nagoshi, Djima Koffi.

**Project administration:** Rodney N. Nagoshi.

**Resources:** Djima Koffi.

**Supervision:** Djima Koffi.

**Writing – original draft:** Rodney N. Nagoshi.

**Writing – review & editing:** Rodney N. Nagoshi, Djima Koffi, Robert L. Meagher, Georg Goergen.

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
