## [Decision Letter · Decision Letter 0]

6 May 2021

PONE-D-21-10287

The fall armyworm strain associated with most rice, millet, and pasture infestations in the Western Hemisphere is rare or absent in Ghana and Togo.

PLOS ONE

Dear Dr. Nagoshi,

Thank you for submitting your manuscript to PLOS ONE. After careful consideration, we feel that it has merit but does not fully meet PLOS ONE’s publication criteria as it currently stands. Therefore, we invite you to submit a revised version of the manuscript that addresses the points raised during the review process.

We look forward to receiving your revised manuscript.

Kind regards,

Ramzi Mansour

Academic Editor

PLOS ONE

Journal Requirements:

3. We note that Figure 1 in your submission contains map images which may be copyrighted.

We require you to either (a) present written permission from the copyright holder to publish this figure specifically under the CC BY 4.0 license, or (b) remove the figure from your submission:

b. If you are unable to obtain permission from the original copyright holder to publish this figure under the CC BY 4.0 license or if the copyright holder’s requirements are incompatible with the CC BY 4.0 license, please either i) remove the figure or ii) supply a replacement figure that complies with the CC BY 4.0 license. Please check copyright information on all replacement figures and update the figure caption with source information. If applicable, please specify in the figure caption text when a figure is similar but not identical to the original image and is therefore for illustrative purposes only.

Reviewers' comments:

Reviewer's Responses to Questions

**Comments to the Author**

1. Is the manuscript technically sound, and do the data support the conclusions?

Reviewer #1: Partly

Reviewer #2: No

2. Has the statistical analysis been performed appropriately and rigorously? 

Reviewer #1: Yes

Reviewer #2: No

3. Have the authors made all data underlying the findings in their manuscript fully available?

Reviewer #1: Yes

Reviewer #2: Yes

4. Is the manuscript presented in an intelligible fashion and written in standard English?

Reviewer #1: Yes

Reviewer #2: Yes

5. Review Comments to the Author

Reviewer #1: Nagoshi et al. analyzed marker sequences (mt-CO1 and TPI) from the samples collected both from rice and corn fields in the fall amryworm and concluded that rice strains do not exist or rarely exist in West Africa. I believe that their argument is supported by their result. However, I have the following concerns.

1. As the authors already showed in their previous papers, the vast majority of fall armyworms in Africa are hybrids between corn and rice strains. A genomic analysis does show evidence that invasive populations are a mixture of corn and rice strains (https://onlinelibrary.wiley.com/doi/full/10.1111/1755-0998.13219). Thus, what we know now is that fall armyworms from corn fiend in invasive populations are hybrids, not corn strain.

Thus, if similar sequences were observed between samples from corn and rice fields, it is fair to say that the samples from rice fields are hybrids, not corn strain.

But throughout the whole manuscript, it appears that they assume that African fall armyworms from corn fields are corn strains. But this assumption seems to be incorrect.

2. In the same context, they have to make it clear what they mean corn strain and rice strain. Are they a taxonomic group? Or just grouping according to the sampling site?

3. Line 51-52. Independent segregation between CO1 and TPI is not true. A half of nuclear genomes is from mother. Thus, 50% of genomes co-segregate with mt-CO1.

4. Table 1, to me, larvae collected from plants by hand could be very different from adults collected by pheromone trap. The adults collected near rice field might have grown up in corn field. They need to show that these adults were grown up on rice field in order not to say that corn strains accidentally arrived near rice fields.

5. L229-L230. They have to provide a reference. And it is really unclear because, at L23-L25, they say that fall armyworms are not a major risk for rice.

6. L347. Indeed, they analyzed a very low number of samples from rice field. Thus, all statistical analyses could not be performed properly because of insufficient statistical power. I do understand that it could have been difficult to collect insects from rice field. And this fact itself might be sufficient to say that there is no established population on rice field.

Reviewer #2: In this paper, authors described “The fall armyworm strain associated with most rice, millet, and pasture infestations in the Western Hemisphere is rare or absent in Ghana and Togo.” Authors have field detection and lab hybrid work upon FAW. Data providing was sufficiently enough, while the inconsistent or conflict recognition on R-strain or C-strain confuse their result and the following discussion.

Actually, authors have a conflict result upon the FAW recognition of R-strain and C-strain between the mitochondrial and nuclear DNA markers, i.e. intermediate types of COI-CS/TpiC and COI-RS/TpiC. They also mentioned and suggested that the COI gene, is not an accurate indicator of strain identity for the African fall armyworm populations. Therefore, how do they elucidate definitely the R-strain is rare or absent in Africa.

Several molecular evidences have shown that R- and C-strains with undistinguishable morphology are generally sympatric in the field, implying the possibility of genetic exchange between the two evolutionary groups. Moreover, molecular evidences also revealed that more than two genetically distinct clusters existed in FAW; therefore, distinguishing the C-strain from Rice-form based on their host plant and genetic markers is unlikely necessary. Authors also shown African FAW do not display strain-specific mating patterns.

It is undoubtedly important work to know the population structure upon African FAW, but it documented unsuitable based on R- and C-strain.

6. PLOS authors have the option to publish the peer review history of their article (what does this mean?). If published, this will include your full peer review and any attached files.

Reviewer #1: No

Reviewer #2: No

---

## [Author Response · Author response to Decision Letter 0]

15 May 2021

Response to Journal.

< 3. We note that Figure 1 in your submission contains map images which may be copyrighted.>

Added to Figure legend that the map image is QGIS open source.

Response to reviewers.

We thank the reviewers for their comments and have tried to address all the concerns. Extensive changes were made to the introduction and discussion to provide a clearer explanation of the strains and markers, as well as the rationale for the experiments. Minor alterations were made to Figures 5 and 6 that did not change the conclusions made from the data. Reviewer’s comments are bracketed (< >).

<1. As the authors already showed in their previous papers, the vast majority of fall armyworms in Africa are hybrids between corn and rice strains… Thus, what we know now is that fall armyworms from corn field in invasive populations are hybrids, not corn strain.>

The assertion that the Africa fall armyworm are hybrids is an explanation for their unusual configuration of genetic markers. However, there is no indication that they are hybrid in their strain identity (in other words as far as we can tell the African fall armyworm are behaving like the C-strain in their plant host use). A paragraph was added to the introduction (lines 71-79) to explain this distinction.

<But throughout the whole manuscript, it appears that they assume that African fall armyworms from corn fields are corn strains. But this assumption seems to be incorrect.>

We have gone through the manuscript to remove such assumptions. For example, the term C-strain or corn strain are not used in the Results section. Instead, the specimens are described by their COI or Tpi haplotype. We also now provide in the discussion an explanation of why we believe the African fall armyworms can be considered C-strain (lines 375-385)

<2. In the same context, they have to make it clear what they mean corn strain and rice strain. Are they a taxonomic group? Or just grouping according to the sampling site?

We added a paragraph to the introduction describing the difficulties in categorizing what the strains represent (lines 47-55). In addition, we note that lines 56-70 in the introduction describe method and rationale for how we defined strains in this paper. 

<3. Line 51-52. Independent segregation between CO1 and TPI is not true. A half of nuclear genomes is from mother. Thus, 50% of genomes co-segregate with mt-CO1.>

We corrected this error in lines 60-62.

<4. Table 1, to me, larvae collected from plants by hand could be very different from adults collected by pheromone trap. The adults collected near rice field might have grown up in corn field. They need to show that these adults were grown up on rice field in order not to say that corn strains accidentally arrived near rice fields.>

As noted in lines 184-190, the larval collections are treated differently from pheromone trap collections when appropriate. We discuss at some length the possibility that the rice pheromone trap specimens come from other hosts (lines 360-372) and have tempered our conclusions with that in mind. We added references to examples where pheromone traps in the Western Hemisphere have successfully collected populations with strain identities that correspond with the local host plants (lines 354-355). 

<5. L229-L230. They have to provide a reference. And it is really unclear because, at L23-L25, they say that fall armyworms are not a major risk for rice.>

We don’t believe a reference here is unnecessary since in this section we describe in detail fall armyworm infestations of rice in South America (references are in 243-247). We also removed lines 23-25 from the revised abstract. 

<6. L347. Indeed, they analyzed a very low number of samples from rice field. Thus, all statistical analyses could not be performed properly because of insufficient statistical power. I do understand that it could have been difficult to collect insects from rice field. And this fact itself might be sufficient to say that there is no established population on rice field.>

We do not understand this criticism. If we look at Figure 5 for example the sample size for rice ranged from 45 to 109, which is pretty high for this type of study and similar in range to the sampling from corn. 

Reviewer #2: In this paper, authors described “The fall armyworm strain associated with most rice, millet, and pasture infestations in the Western Hemisphere is rare or absent in Ghana and Togo.” Authors have field detection and lab hybrid work upon FAW. Data providing was sufficiently enough, while the inconsistent or conflict recognition on R-strain or C-strain confuse their result and the following discussion.>

We made major revisions to the introduction and discussion to address this criticism.

<Actually, authors have a conflict result upon the FAW recognition of R-strain and C-strain between the mitochondrial and nuclear DNA markers, i.e. intermediate types of COI-CS/TpiC and COI-RS/TpiC. They also mentioned and suggested that the COI gene, is not an accurate indicator of strain identity for the African fall armyworm populations. Therefore, how do they elucidate definitely the R-strain is rare or absent in Africa.>

The conclusion that the R-strain is rare is based on the frequency of the TpiR marker and lack of fall armyworm in host plants and habitats preferred by the R-strain. In response to the reviewer, we expanded the relevant section of the Discussion on this point (lines 349-374). We also describe why we believe that the Tpi marker may still be relevant for strain identification in the introduction (lines 71-79) and Discussion (lines 375-399).

<Several molecular evidences have shown that R- and C-strains with undistinguishable morphology are generally sympatric in the field, implying the possibility of genetic exchange between the two evolutionary groups. Moreover, molecular evidences also revealed that more than two genetically distinct clusters existed in FAW; therefore, distinguishing the C-strain from Rice-form based on their host plant and genetic markers is unlikely necessary. >

Not sure what other genetic clusters are being referred to here, but we believe the strains are of particular importance because they differentiate fall armyworm based on plant host use. This has important implication when determining what plant types are at risk. We added a section to better explain this point of view to the introduction (lines 71-86). We also added additional material to the discussion describing our interpretation of the fall armyworm strains found in Africa (lines 375-399).

<Authors also shown African FAW do not display strain-specific mating patterns.

It is undoubtedly important work to know the population structure upon African FAW, but it documented unsuitable based on R- and C-strain.>

We disagree. At this time the strain markers are the only ones that have been demonstrated (in the Americas) to distinguish fall armyworm for their host plant preference, which has obvious relevance to identifying what crops are at risk. As detailed above, we have made extensive revisions to the introduction and discussion to explain our perspective.

---

## [Decision Letter · Decision Letter 1]

27 May 2021

PONE-D-21-10287R1

The fall armyworm strain associated with most rice, millet, and pasture infestations in the Western Hemisphere is rare or absent in Ghana and Togo.

PLOS ONE

Dear Dr. Nagoshi,

Thank you for submitting your manuscript to PLOS ONE. After careful consideration, we feel that it has merit but does not fully meet PLOS ONE’s publication criteria as it currently stands. Therefore, we invite you to submit a revised version of the manuscript that addresses the points raised during the review process.

We look forward to receiving your revised manuscript.

Kind regards,

Ramzi Mansour

Academic Editor

PLOS ONE

Reviewers' comments:

Reviewer's Responses to Questions

**Comments to the Author**

1. If the authors have adequately addressed your comments raised in a previous round of review and you feel that this manuscript is now acceptable for publication, you may indicate that here to bypass the “Comments to the Author” section, enter your conflict of interest statement in the “Confidential to Editor” section, and submit your "Accept" recommendation.

Reviewer #2: (No Response)

2. Is the manuscript technically sound, and do the data support the conclusions?

Reviewer #2: Partly

3. Has the statistical analysis been performed appropriately and rigorously? 

Reviewer #2: N/A

4. Have the authors made all data underlying the findings in their manuscript fully available?

Reviewer #2: Yes

5. Is the manuscript presented in an intelligible fashion and written in standard English?

Reviewer #2: Yes

6. Review Comments to the Author

Reviewer #2: Authors could have an outstanding elucidation for African FAW structure, while the R-strain and C-strain is now not a natural recognition.

As the authors showed either in the previous version or this version, L71-79, that hybridization between Corn-strain and Rice-strain has been occurred and showed disagreement between the mtCOI and nrTpi strain haplotypes. Authors also know that there have several publishings documented that more than two genetic lineages could be found in FAW. It is already unmeaningful to distinguish R- from C-strains for FAW as I have suggested in the previous version. It is undoubtedly important work to know the population structure upon African FAW, but the R- and C-strain documentation should be abandoned.

7. PLOS authors have the option to publish the peer review history of their article (what does this mean?). If published, this will include your full peer review and any attached files.

Reviewer #2: No

---

## [Author Response · Author response to Decision Letter 1]

29 May 2021

We do not agree with the reviewer’s comments.

The primary objection of the reviewer is: “It is already unmeaningful to distinguish R- from C-strains for FAW as I have suggested in the previous version”. We strongly disagree with that statement and believe the reviewer has not taken into consideration the following points. 

First, all previous work on FAW population genetics in Africa have been done on specimens collected from corn or less frequently sorghum. These are host plants associated with the C-strain. Very little is known about whether FAW is associated with R-strain hosts in Africa and if so, what the strain composition of these might be. I want to emphasize that point, the characterization of FAW associated with R-strain host plants in Africa has not been done previously and without that analysis one cannot assume the absence of the R-strain in Africa. To do so, as appears to be what is being suggested by the reviewer, is not good science. This point is explicitly made several times in the paper (see for example lines 8-16 in the abstract, lines 85-89 in the introduction, and lines 353-354 in the Discussion). It is even alluded to in the lines noted by the reviewer (L71-79) where we state “Because virtually all the Eastern Hemisphere specimens tested to date were collected from the C-strain associated hosts corn and sorghum,…” (L74-76).

This is the issue being addressed by this manuscript. This is the first, and so far only, attempt to systematically collect and genetically analyze FAW in Africa from R-strain hosts. As such we believe it is a critically important paper for assessing the status of the two strains in Africa. To make this point explicitly clear lines 79-81 and 88-89 were added to the Introduction.

Second, the assertion by the reviewer that the strains are no longer relevant in Africa and that 

 “the R- and C-strain documentation should be abandoned” is at best premature and most likely wrong. The defining characteristic of the strains is host plant preference. The defining characteristic of the C-strain is that it primarily infests corn and sorghum fields, only occasionally rice, and very rarely pasture grasses. This manuscript demonstrates that the Africa FAW is showing a similar preference pattern. It is found in corn, is ambiguous in rice (trap captures but little evidence of larval infestation), and virtually absent in pasture grasses. So even if there was an ancestral hybridization event between the strains, the Africa FAW population is effectively still behaving like the C-strain. What has changed is that COI is no longer an accurate strain marker. This point is explicitly made in Lines 74-78 in the Introduction and 384-387 in the Discussion. We also provided an explanation for how this might have occurred where an early hybridization event between strains disassociated the COI marker but still maintained a C-strain identity/behavior (lines 377-401).

Third, the reviewer does not consider the fact that the R-strain does exist in the Western Hemisphere and so could enter Africa at any time. So even if the current African FAW is a hybrid, secondary introductions of FAW to Africa could return both strains into the continent with potential economic consequences. Therefore, continued surveillance of Africa populations with the strain markers is still relevant and important as is noted in lines 421-424.

The reviewer also stated: “Authors also know that there have several publishings (sic) documented that more than two genetic lineages could be found in FAW.” We believe these lineages are irrelevant to this paper as the lineages are not associated with host preference. This paper is about whether the two FAW lineages associated with different host plant preferences in the Western Hemisphere are present in Africa.

In summary, we believe the reviewer’s objections are not reasonable and stem from assumptions that are premature and probably incorrect. This manuscript represents the first systematic attempt to find and genetically characterize Africa FAW in the R-strain hosts of rice and pasture grasses. This information is critical to the assessment as to whether the fall armyworm population likely to threaten rice, millet, and pasture grasses is present Africa.

---

## [Editor Report · Decision Letter 2]

8 Jun 2021

The fall armyworm strain associated with most rice, millet, and pasture infestations in the Western Hemisphere is rare or absent in Ghana and Togo.

PONE-D-21-10287R2

Dear Dr. Nagoshi,

We’re pleased to inform you that your manuscript has been judged scientifically suitable for publication (BUT, please see and apply ADDITIONAL EDITOR COMMENTS below) and will be formally accepted for publication once it meets all outstanding technical requirements.

Kind regards,

Ramzi Mansour

Academic Editor

PLOS ONE

Additional Editor Comments:

The following revisions should be made by the authors on the PROOFS of their accepted article:

L23 (Abstract):  please replace "Ghana-Togo region"   with   "Ghana-Togo area"

L27 (Introduction):  please replace  "(*Spodoptera frugiperda*) is"    with    "*Spodoptera frugiperda* (J.E. Smith) (Lepidoptera: Noctuidae) is"

L34:  please replace "plant hosts"  with  "host plants"

L54:  please replace "and plant host"   with  "and host plant"

L100 (Materials and methods):  please replace "Fall armyworm was collected"   with   "Fall armyworm individuals were collected"              

L124:  please leave one space after "6000"

L124:  please delete the  " . "   after   "5 min"

L130:  10 mM

L145:  please add a comma before  "the first PCR"

L185-186:   the number of TpiH specimens were incorporated ??   something is wrong here, please correct (the numbers ?)

L238 (Figure 3 caption):  please replace  "pheromone trap capture numbers"   with  "pheromone trap captures"

L259:  please replace  "plant types"   with   "plant species"

L263:  please add  "and"   before  "Murua"

L298:  please add the Authorship, Order and Family  "(Boisduval) (Lepidoptera: Noctuidae)"    just after  "*Spodoptera littorali*s"  as this species is mentioned for the first time here

L299:  the common name abbreviation "FAW" is mentioned for the first time in the paper with no previous definition; for consistency with what is written throughout the text, I'd suggest replacing "FAW"  with  "fall armyworm"

L301:  Larvae were also

L327:  please leave one space before  "(Fig 6)"

L352 (Discussion):  please replace  "affected regions"    with   "affected areas"

L358:  please add a comma after  "efforts"

L364:  please add a comma after "and Togo"

L365-366:   please replace  "in this region"  with  "in this western African area"

L368:  please change to ", that fall armyworms collected in the"

L373-373:  please change to  "Regardless of origin, fall armyworms found in rice traps were"

L419:  please replace  "Africa populations"   with  "African fall armyworm populations"

L420:  associated host plants
---

## [Editor Report · Acceptance letter]

11 Jun 2021

PONE-D-21-10287R2 

The fall armyworm strain associated with most rice, millet, and pasture infestations in the Western Hemisphere is rare or absent in Ghana and Togo. 

Dear Dr. Nagoshi:

I'm pleased to inform you that your manuscript has been deemed suitable for publication in PLOS ONE. Congratulations! Your manuscript is now with our production department. 

Kind regards, 

on behalf of

Dr. Ramzi Mansour 

Academic Editor

PLOS ONE